# Adversarial Learning for Robust Deep Clustering

**Xu Yang[1]    Cheng Deng[1]***    **Kun Wei[1]    Junchi Yan[2]    Wei Liu[3]**
[1]School of Electronic Engineering, Xidian University, Xian 710071, China
[2]Department of CSE and MoE Key Lab of Artificial Intelligence, Shanghai Jiao Tong University
[3]Tencent AI Lab, Shenzhen, China
{xuyang.xd, chdeng.xd, weikunsk}@gmail.com, yanjunchi@sjtu.edu.cn, wl2223@columbia.edu

## Abstract

Deep clustering integrates embedding and clustering together to obtain the optimal nonlinear embedding space, which is more effective in real-world scenarios compared with conventional clustering methods. However, the robustness of the clustering network is prone to being attenuated especially when it encounters an adversarial attack. A small perturbation in the embedding space will lead to diverse clustering results since the labels are absent. In this paper, we propose a robust deep clustering method based on adversarial learning. Specifically, we first attempt to define adversarial samples in the embedding space for the clustering network. Meanwhile, we devise an adversarial attack strategy to explore samples that easily fool the clustering layers but do not impact the performance of the deep embedding. We then provide a simple yet efficient defense algorithm to improve the robustness of the clustering network. Experimental results on two popular datasets show that the proposed adversarial learning method can significantly enhance the robustness and further improve the overall clustering performance. Particularly, the proposed method is generally applicable to multiple existing clustering frameworks to boost their robustness. The source code is available at https://github.com/xdxuyang/ALRDC.

## 1   Introduction

As an important tool in unsupervised learning [38], clustering has been widely utilized in image segmentation [8], image categorization [10], and data mining and analysis. The goal of clustering is to find a partition to keep similar samples in the same cluster while dissimilar ones in different clusters. Recently, a large family of clustering algorithms such as $K$-means clustering [12] and Gaussian mixture models [1] have been studied intensively. $K$-means clustering assigns each sample to the cluster with the closest center iteratively based on similarity measurements and updates the center of each cluster. However, the estimated similarity measures for high-dimensional samples may not be accurate, resulting in degraded clustering performances. In practice, many high-dimensional samples may exhibit a dense grouping property in a low-dimensional representation space. Hence, clustering methods, such as spectral clustering [26] and subspace clustering [7], have been developed to capture various cluster structures.

A majority of spectral clustering approaches [36] depend on a linear subspace assumption to construct affinity matrices, but data does not naturally obey linear models in many cases. Their distance metrics are only exploited to describe local relationships in the data space, and have a limitation to represent latent dependencies among all samples. Moreover, these shallow clustering methods mainly rely on low-level image features such as raw pixels, SIFT [23], or HOG [6]. On the contrary, deep clustering, which integrates embedding and clustering as a single procedure to obtain the optimal embedding

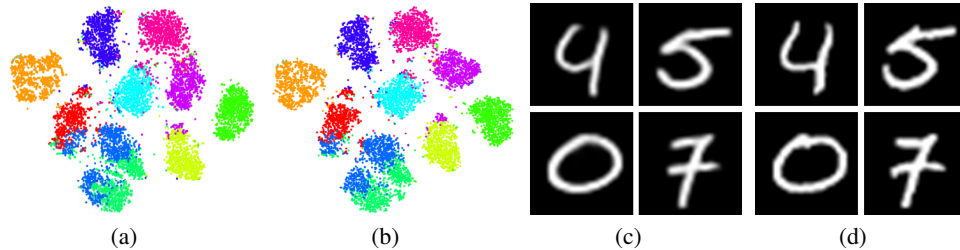

| (a) | (b) | (c) | (d) |

Figure 1: Visualization of the discriminative capability of embedding spaces. (a) The distribution of the embedded features on MNIST-*test* and their clustering results (ACC=0.849); (b) the distribution of the embedded features with a small perturbation on MNIST-*test* and their clustering results (ACC=0.772); (c) and (d) are the reconstructed samples generated by the embedded features and their perturbed versions.

(or representation) space for clustering, can be more effective than traditional methods. The main reason is due to the powerful ability of deep methods in effectively modeling the distributions of input samples [21] and capturing the nonlinear properties [35, 30], which is more suitable for real-world clustering scenarios [20].

Deep autoencoder network [29] has promoted the performance of deep clustering due to its inherent property to capture high-dimensional probability distributions of input samples without label information. The encoder embeds the samples into a latent lower-dimensional space, and adopts an explicit approximation of maximum likelihood to estimate the distribution diversity between the embedded features and the raw samples. The embedded features are utilized to produce the final clustering results. They can also reconstruct the raw samples by the decoder network, which is leveraged to ensure that the embedded features preserve the information of the raw samples [28]. More recently, Deep Embedding Clustering [32] has been proposed to train an end-to-end clustering network, which learns a mapping from the data space to a lower-dimensional embedding space. In order to benefit from end-to-end optimization and also eliminate the necessity of layer-wise pre-training, joint learning frameworks [9, 34], which propose to minimize unified clustering and reconstruction loss functions, train all the network layers simultaneously. While deep learning algorithms have been extremely successful at a variety of learning tasks, it has been shown that deep neural networks can be easily fooled by adversarially generated samples [24]. An adversarial sample is generally indistinguishable from its original sample by a human observer and is misclassified by neural networks. Since then, there have been a lot of researches undertaken to make supervised models resilient to adversarial samples, and to expose vulnerabilities in existing deep learning algorithms [3].

Such a kind of vulnerabilities also exist in unsupervised deep clustering networks. The majority of existing deep clustering methods endeavor to minimize the reconstruction loss, and their goal is to make the target embedding space more discriminative since the embedding space directly determines the clustering quality. However, the embedded features are extremely susceptible to a small perturbation and lead to disparate clustering results. For example, we jointly optimize an autoencoder network and clustering layers with $KL$ divergence as an end-to-end clustering network. Figure 1 gives the performance by comparing different embedded features on MNIST-*test*, where Figures 1(a) and 1(b) plot the distributions of the embedded features and their perturbed versions, respectively, using $t$-SNE visualization [19]. Different colors indicate their corresponding clusters. Moreover, the reconstructed samples with different clustering results from the embedded features and their perturbed versions are displayed in Figures 1(c) and 1(d), respectively. The results show that a small perturbation will cause the clustering results quite different, and the reconstruction loss used by the autoencoder network cannot sufficiently perceive the adversarial perturbation.

In this paper, we first introduce an adversarial learning algorithm to improve the network robustness in deep clustering. We attempt to define an adversarial sample of the embedding space, which easily fools the clustering layers but does not impact the performance of the deep embedding. Meanwhile, we present a powerful adversarial attack algorithm to learn a small perturbation from the embedding space against the clustering network. In this way, those unstable samples that are very likely to yield diverse clustering results are explored explicitly. Moreover, we provide a simple yet efficient defense algorithm to optimize the clustering network, which can alleviate the differences caused by the perturbation. Experimental results on two popular benchmark datasets show that the proposed

adversarial learning algorithm can significantly enhance the robustness and further improve the overall performance of the clustering network. Our proposed method can be integrated into multiple classic unsupervised clustering frameworks to enhance their robustness.

**Contributions.** The highlights of this paper are three-fold: 1) We first attempt to optimize the clustering network using an adversarial learning mechanism, and define an adversarial sample for the embedding space of deep clustering. 2) We present a powerful adversarial attack strategy against the clustering network to explore unstable samples, and propose a corresponding defense algorithm which improves the overall clustering performance while boosts the robustness of the network. 3) The experimental results on two popular datasets show that the proposed adversarial learning algorithm can optimize the feature distribution to alleviate the effect caused by a perturbation, therefore enhancing the robustness of various existing clustering frameworks.

## 2 Background and Preliminaries

**Notations.** Let $\mathbf{X} = \{\mathbf{x}_1, ..., \mathbf{x}_n\}$ be input samples, and $\mathbf{Z} = \{\mathbf{z}_1, ..., \mathbf{z}_n\}$ be their embedded features, respectively. $\mathbf{z}_i \in \mathbb{R}^d$ is learned by the embedding network $E$, and on the other hand it is utilized to reconstruct the raw sample $\mathbf{x}_i$. We use a clustering function $F: \mathbf{z} \to \mathbf{y} \in \mathbb{R}^K$ to predict the cluster label, where $K$ is the total number of clusters, and then let $\mathbf{Y} = \{\mathbf{y}_1, ..., \mathbf{y}_n\}$ retain the final clustering results.

Our adversarial learning algorithm is model-agnostic and can therefore be applied to any deep clustering model that follows the $\mathbf{x} \rightleftarrows \mathbf{z} \to \mathbf{y}$ structure. Hence, we adopt the following clustering network, which is a basic model of an existing method [37], as a testbed to show how our proposed adversarial learning algorithm can attack the network and improve its robustness. The ultimate clustering network combines embedding and clustering as a whole to produce the optimal nonlinear embedding. Two modules are merged into one unified framework and jointly optimized by relative-entropy (equivalent to $KL$ divergence) minimization, and the loss function can be defined as:

$$\min_{\Theta} \ \mathcal{L}_C = KL(p(\mathbf{x}, \mathbf{z}, \mathbf{y}) || q(\mathbf{x}, \mathbf{z}, \mathbf{y})). \tag{1}$$

In order to solve the above problem, we introduce the following generative model:

$$\begin{aligned} \mathbf{y} &\sim Cat(\boldsymbol{\pi}), \\ \mathbf{z} &\sim \mathcal{N}(\mu_z(\mathbf{y}), \sigma_z^2(\mathbf{y})), \\ \mathbf{x} &\sim \text{Gaussian}(\mu_x(\mathbf{z})), \end{aligned} \tag{2}$$

with the joint probabilities factorized as $p(\mathbf{z}, \mathbf{y}|\mathbf{x}) = p(\mathbf{y}|\mathbf{z})p(\mathbf{z}|\mathbf{x})$, $q(\mathbf{x}|\mathbf{z}, \mathbf{y}) = q(\mathbf{x}|\mathbf{z})$, and $q(\mathbf{z}, \mathbf{y}) = q(\mathbf{z}|\mathbf{y})q(\mathbf{y})$. The loss of the clustering network becomes:

$$\min_{\Theta} \mathcal{L}_C = \mathbb{E}_{\mathbf{x} \sim p_{data}(\mathbf{x})} \left[ -\log q(\mathbf{x}|\mathbf{z}) + \sum_{\mathbf{y}} p(\mathbf{y}|\mathbf{z}) \log \frac{p(\mathbf{z}|\mathbf{x})}{q(\mathbf{z}|\mathbf{y})} + KL(p(\mathbf{y}|\mathbf{z})||q(\mathbf{y})) \right], \mathbf{z} \sim p(\mathbf{z}|\mathbf{x}), \tag{3}$$

where the first term is the reconstruction loss. In a deep clustering scenario, $\mathbf{y}$ can be interpreted as representing some discrete clusters in the data, and $\mathbf{z}$ then represents a mixture of Gaussians, encoding both the inter- and intra-cluster variations. In this way, the raw samples are encoded to an embedding space, and the mixture probabilities $p(\mathbf{y}|\mathbf{z})$ are determined by an output Softmax operator of the clustering layers. Finally, the decoder is a single network that maps from the embedding $\mathbf{z}$ to its reconstructed sample $\tilde{\mathbf{x}}$. The last term is the categorical regularizer. $q(\mathbf{y})$ is a fixed uniform prior with component weights specified by $\boldsymbol{\pi}$. Intuitively, this loss encourages the model to reconstruct the raw samples and perform clustering where possible. Due to the space limit, please refer to our supplementary material for the detailed formulations.

By exploiting the diversity in input samples $\mathbf{X}$, the model can learn to utilize different components for different structures inherent in the data. Then, we can directly infer the cluster labels through the clustering layers. In specific, $s_i \in \{1, ..., K\}$ is the inferred cluster label of input sample $\mathbf{x}_i$:

$$s_i = \text{argmax } \mathbf{y}_i. \tag{4}$$

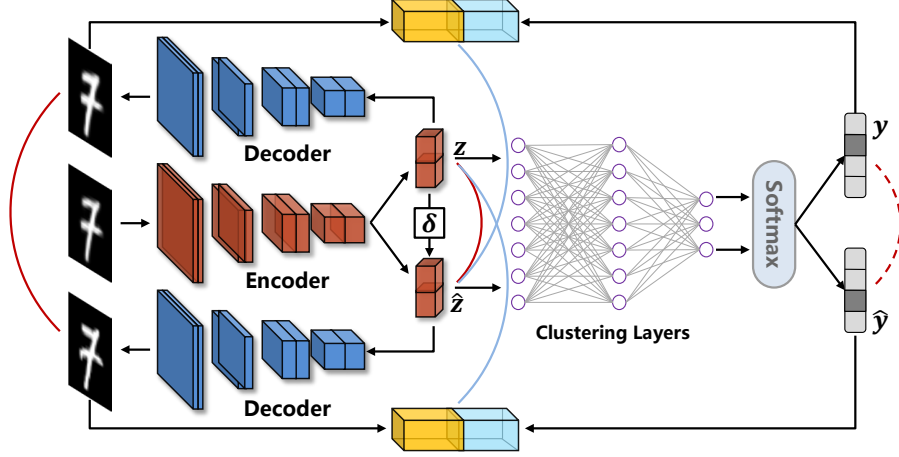

Figure 2: The illustration of the entire architecture of our proposed method.

## 3 Methodology

We leverage the deep clustering network introduced in Section 2 as a testbed to show the process of the proposed adversarial learning algorithm. The overall flowchart is illustrated in Figure 2, where red lines represent the adversarial attack strategy and blue lines stand for the defense strategy. For a pre-trained deep clustering network (with parameters $\Theta$) that consists of an embedding subnetwork and clustering layers, we intend to generate a small perturbation $\boldsymbol{\delta}$ via an attacking network (with parameters $\Phi$) for the embedded feature $\mathbf{z}$, such that an adversarial sample in the embedding space can be defined as $\hat{\mathbf{z}} = \mathbf{z} + \boldsymbol{\delta}$. In this way, feeding $\hat{\mathbf{z}}$ into the target clustering network, the reconstructed sample $\hat{\mathbf{x}}$ and its clustering label $\hat{\mathbf{y}}$ can be yielded as follows,

$$\hat{\mathbf{y}} \sim p(\mathbf{y}|\hat{\mathbf{z}}), \hat{\mathbf{x}} \sim q(\mathbf{x}|\hat{\mathbf{z}}). \tag{5}$$

The purpose of adversarial attack is to make the perturbed features similar to the clean features and the reconstructed samples have fewer differences, but make the corresponding clustering results quite different. The loss of the adversarial attack learning can be defined as:

$$\min_{\Phi} \ \mathcal{L}_A = \| \ \tilde{\mathbf{x}} - \hat{\mathbf{x}} \ \|_{\mathrm{F}}^2 + \beta \|\boldsymbol{\delta}\|_p + \gamma \sum_{i=1}^{n} \mathbf{y}_i^\top \hat{\mathbf{y}}_i, \tag{6}$$

where the first term is utilized to reconstruct samples with the perturbed features, the second term ensures that the learned perturbation will not destroy the basic performance of the clustering network, and the last term is to maximize the differences in clustering results. $\beta$ and $\gamma$ are trade-off hyper-parameters. To solve the above problem, we fix the network parameters $\Theta$ and optimize $\Phi$. In this way, a small perturbation is learned to fool the clustering layers but not impact the performance of the deep embedding. This is because the cluster structure of some samples is not clear in the embedding space. By virtue of the adversarial attack strategy, the unclear samples which are very likely to cause diverse clustering results are explored explicitly.

After learning the perturbation, we hope to use the learned perturbation in defense of the clustering network. The generated samples with the perturbed features are basically the same as those generated by the clean features, while the subsequent clustering results are quite different. However, the ideal situation is that the generated samples as well as the clustering results by the perturbed features should be completely consistent with those by the clean features. Hence, we combine the clustering results and the reconstructed samples to obtain new feature maps, and adopt a discriminator (with parameters $\Psi$) to identify the mutual information between $(\mathbf{x}, \mathbf{y})$ and $\mathbf{z}$. The generated samples and their corresponding cluster labels are combined via feature reshaping:

$$\min_{\Psi} \ \mathcal{L} = \mathbb{E}_{\mathbf{x} \sim p_{data}(\mathbf{x})} \left[ -\log \sigma(T((\mathbf{x}, \mathbf{y}), \mathbf{z})) - \log(1 - \sigma(T((\mathbf{x}_t, \mathbf{y}_t), \mathbf{z}))) \right], \tag{7}$$

where $((\mathbf{x}, \mathbf{y}), \mathbf{z})$ together forms a positive data pair, and then we randomly select $(\mathbf{x}_t, \mathbf{y}_t)$ from the disturbed batch to construct a negative data pair with respect to $\mathbf{z}$. Notation $T$ denotes the mapping

---

**Algorithm 1**   Adversarial Learning for Robust Deep Clustering

---

**Input**: Unlabeled samples $\mathbf{X}$, cluster number $K$, hyper-parameters $\lambda$, $\beta$, and $\gamma$;
**Initialization**: Pre-train the clustering network by minimizing Eq. (3);
**Output**: $\mathbf{Y}$;

  1: **for** $iter = 1, \ldots, MaxIter$ **do**
  2:    **while** not converged **do**
  3:        Fixing $\Psi$ and $\Theta$, update the network parameters $\Phi$ by minimizing Eq. (6);
  4:    **end while**
  5:    **while** not converged **do**
  6:        Fixing $\Phi$, update the network parameters $\Psi$ and $\Theta$ by minimizing Eq. (8);
  7:    **end while**
  8: **end for**

---

function of the discriminator which is often used to discriminate the mutual information of inputs [13], and $\sigma$ represents the activation function of the discriminator. In doing so, the correlation between $\mathbf{z}$ and $(\mathbf{x}, \mathbf{y})$ will be enhanced. Moreover, contrastive learning [4] is utilized to force the results generated by the clean features and their perturbed versions to be similar. Specifically, we employ the discriminator to optimize the correlation between $\hat{\mathbf{z}}$ and $(\tilde{\mathbf{x}}, \mathbf{y})$, and the negative data pair $(\tilde{\mathbf{x}}_t, \mathbf{y}_t)$ is randomly selected from the disturbed batch. Similarly, the correlation between $\mathbf{z}$ and $(\hat{\mathbf{x}}, \hat{\mathbf{y}})$ is constructed. The loss of the defense strategy can be defined as:

$$
\begin{aligned}
\min_{\Psi, \Theta} \mathcal{L}_D = {}& \mathbb{E}_{\mathbf{x} \sim p_{data}(\mathbf{x})}[- \log \sigma(T((\hat{\mathbf{x}}, \hat{\mathbf{y}}), \mathbf{z})) - \log(1 - \sigma(T((\hat{\mathbf{x}}_t, \hat{\mathbf{y}}_t), \mathbf{z})))] \\
& + \mathbb{E}_{\mathbf{x} \sim p_{data}(\mathbf{x})}[- \log \sigma(T((\tilde{\mathbf{x}}, \mathbf{y}), \hat{\mathbf{z}})) - \log(1 - \sigma(T((\tilde{\mathbf{x}}_t, \mathbf{y}_t), \hat{\mathbf{z}})))] + \lambda \mathcal{L}_C.
\end{aligned}
\tag{8}
$$

We fix the network parameters $\Phi$ and optimize $\Psi, \Theta$ to improve the network robustness. $\mathcal{L}_C$ is the objective function of the original deep clustering method, which is included to ensure the basic performance of the clustering network. Particularly, the clustering network introduced in Section 2 is one of such models leveraged to introduce an adversarial learning algorithm and afterwards verify its effectiveness encountering an attack algorithm. $\mathcal{L}_C$ changes as the clustering model changes. To be precise, our defense algorithm is to integrate a set of perturbation-based contrastive constraints into the original clustering network, which can force the embedded features away from the decision boundaries of the clusters to eliminate the different results caused by the learned perturbation. As such, the robustness and overall performance of the clustering network will both be improved. Finally, our proposed adversarial learning algorithm for robust deep clustering is sketched in Algorithm 1.

## 4   Related Works

**Deep Clustering.** A number of related methods aim to learn a discriminative embedding space using generative models. Nalisnick *et al.* [22] adopted a latent mixture of Gaussians based on VAE [16] and proposed a Bayesian non-parametric prior aiming to capture the class structure in an unsupervised manner. Similarly, Joo *et al.* [15] presented a Dirichlet posterior in the embedding space to avoid some of the previously observed component-collapsing phenomena. Lastly, Variational Deep Embedding (VaDE) [14] is proposed to combine VAE and GMM together for deep clustering. Moreover, several works have shown that random noise can be utilized to make the embedded features more robust [37, 9] towards clear cluster structure. The perturbations generated by random noises are irregular and chaotic, and it is thus very hard to explore the samples that are susceptible to perturbations, causing a performance gap which is to be bridged by this work. Our proposed method can utilize a small perturbation to explicitly explore unstable samples in the embedding space.

**Adversarial Learning.** An adversarial setting [27] in clustering is firstly introduced to make mis-clustering using fringe clusters, where adversaries could place adversarial samples very close to the decision boundary of the original data cluster. Biggio *et al.* [2] considered adversarial attack to clustering, where they described the obfuscation and poisoning attack settings, and then provided results on single-linkage hierarchical clustering. Recently, Crussell and Kegelmeyer [5] proposed a poisoning attack specific to DBSCAN clustering. As can be seen, a few works have discussed adversarial learning on deep clustering, and the stability and robustness are also very crucial to deep clustering networks. This paper not only defines an adversarial attack to the embedding space for deep clustering but also presents a defense strategy to improve the network robustness.

# 5 Experiments

In this section, we evaluate the effectiveness of the proposed adversarial learning method on two benchmark datasets in terms of: 1) whether our attack method learns a meaningful perturbation from the embedding space, 2) whether the proposed defense strategy can improve the robustness of the clustering network, and 3) the applicability of the proposed method to other classic clustering frameworks.

## 5.1 Datasets

To show that our method operates well on various datasets, we choose MNIST and Fashion-MNIST as benchmarks. Considering that clustering tasks are fully unsupervised, we concatenate the training and testing samples when applicable. MNIST [18]: containing a total of 70,000 handwritten digits with 60,000 training and 10,000 testing samples, each being a 28×28 monochrome image. Fashion-MNIST [31]: having the same number of images with the same image size as MNIST, but fairly more complicated. Instead of digits, Fashion-MNIST consists of various types of fashion products.

## 5.2 Clustering Metrics

To evaluate the clustering performance, we adopt three standard evaluation metrics: Accuracy (ACC), Normalized Mutual Information (NMI) [33], and Distortion (D).

The best mapping between cluster assignments and true labels is computed using the Hungarian algorithm to measure accuracy [17]. For completeness, we define ACC by:

$$ACC = \max_{m} \frac{\sum_{i=1}^{n} \mathbf{1}\{l_i = m(c_i)\}}{n}, \tag{9}$$

where $l_i$ and $c_i$ are the ground-truth label and predicted cluster label of data point $\mathbf{x}_i$, respectively.

NMI calculates the normalized measure of similarity between two labels of the same data:

$$NMI = \frac{I(l; c)}{max\{H(l), H(c)\}}, \tag{10}$$

where $I(l, c)$ denotes the mutual information between true label $l$ and predicted cluster $c$, and $H$ represents their entropy. Results of NMI do not change by permutations of clusters (classes), and they are normalized to [0, 1] with 0 implying no correlation and 1 exhibiting perfect correlation.

The distortion D between the embedded features and their perturbed versions is measured by $D = \frac{1}{nd} \sum_{i=1}^{n} \frac{|\mathbf{z}_i - \hat{\mathbf{z}}_i|}{|\mathbf{z}_i|}$, where $d$ is the dimension of embedded features.

## 5.3 Implementation Details

In our experiments, we set $\lambda = 1$. The hyper-parameters $\beta$ and $\gamma$ are determined by different networks and datasets. We aim to select the hyper-parameters that can achieve a certain difference in the result (the 3rd term of Eq. (6)) by a moderate perturbation (not too much). The roles of $\beta$ and $\gamma$ are mutually exclusive, so we typically fix $\beta$ and tune $\gamma$. The experiments show that the difference in the result will increase abruptly as $\gamma$ gradually increases, and that the critical value $\gamma$ is an ideal hyper-parameter. For MNIST, the channel numbers and kernel sizes of the autoencoder network are the same as those in [37], and we employ one convolutional layer and three following residual blocks in the encoder for Fashion-MNIST. The clustering layers consist of four fully-connected layers, and ReLU is employed as nonlinear activation. The perturbed clustering results are marked by (*) on top, and the model after the defense strategy is marked by (+) on top.

## 5.4 Baselines

We first verify the robustness and stability of the basic clustering network mentioned in Section 2 (ConvAE in the experiments). In addition, we integrate some classic modules of unsupervised learning with the basic clustering network to verify the applicability of the proposed adversarial algorithm, such as mutual information estimation (MIE) [13] and graph module (Graph) [25]. The specific

Table 1: Clustering performances (%) of different methods after adversarial attack learning on two datasets in ACC, NMI, and D.

| Dataset | Method | Matrix | 64 | | 128 | | 256 | | 512 | |
|---|---|---|---|---|---|---|---|---|---|---|
| MNIST | ConvAE | ACC | 85.7 | 77.5* | 84.9 | 77.2* | 85.8 | 77.5* | 84.4 | 75.3* |
| | | NMI | 80.4 | 75.5* | 79.7 | 76.6* | 81.9 | 77.9* | 79.9 | 72.1* |
| | | D | 1.06 | | 0.73 | | 0.64 | | 0.51 | |
| | MIE | ACC | 90.2 | 81.7* | 91.3 | 83.6* | 92.9 | 82.1* | 91.8 | 82.6* |
| | | NMI | 85.4 | 78.0* | 85.8 | 80.2* | 86.3 | 80.9* | 84.7 | 78.9* |
| | | D | 1.14 | | 0.65 | | 0.56 | | 0.47 | |
| | Graph | ACC | 95.3 | 88.2* | 96.2 | 88.5* | 95.3 | 88.2* | 96.1 | 89.9* |
| | | NMI | 94.5 | 84.3* | 94.7 | 85.2* | 95.1 | 85.7* | 94.5 | 84.4* |
| | | D | 1.02 | | 1.03 | | 1.04 | | 1.24 | |
| Fashion-MNIST | ConvAE | ACC | 60.6 | 56.0* | 61.3 | 57.1* | 61.7 | 56.3* | 60.9 | 54.9* |
| | | NMI | 63.1 | 58.8* | 64.1 | 59.7* | 63.1 | 57.9* | 63.5 | 58.8* |
| | | D | 1.91 | | 1.25 | | 1.18 | | 1.28 | |
| | MIE | ACC | 65.4 | 58.5* | 64.2 | 57.1* | 64.9 | 57.7* | 64.9 | 57.5* |
| | | NMI | 64.9 | 62.6* | 63.8 | 61.2* | 64.2 | 61.7* | 63.9 | 61.6* |
| | | D | 1.30 | | 1.15 | | 1.10 | | 1.05 | |
| | Graph | ACC | 67.2 | 64.3* | 66.9 | 63.4* | 66.7 | 63.8* | 66.8 | 64.0* |
| | | NMI | 66.5 | 63.8* | 66.4 | 63.3* | 66.5 | 63.6* | 66.6 | 63.8* |
| | | D | 1.95 | | 1.93 | | 1.84 | | 1.72 | |

Table 2: Clustering performances (%) of different methods after the adversarial defense strategy on two datasets in ACC and NMI.

| Dataset | Method | Matrix | 64 | | 128 | | 256 | | 512 | |
|---|---|---|---|---|---|---|---|---|---|---|
| MNIST | ConvAE$^+$ | ACC | 86.8 | 87.1* | 87.2 | 87.0* | 86.8 | 87.1* | 86.4 | 86.7* |
| | | NMI | 82.8 | 82.7* | 83.1 | 82.6* | 82.5 | 82.9* | 80.3 | 80.8* |
| | MIE$^+$ | ACC | 93.9 | 94.0* | 94.2 | 94.3* | 94.5 | 94.2* | 94.0 | 94.1* |
| | | NMI | 85.6 | 85.9* | 86.1 | 87.2* | 85.7 | 84.9* | 86.1 | 84.2* |
| | Graph$^+$ | ACC | 98.2 | 98.5* | 98.1 | 97.7 * | 98.3 | 97.5* | 97.5 | 97.9* |
| | | NMI | 94.8 | 94.5* | 94.2 | 93.8* | 94.7 | 94.1* | 94.2 | 93.7* |
| Fashion-MNIST | ConvAE$^+$ | ACC | 61.8 | 62.3* | 63.1 | 62.5* | 62.8 | 63.7* | 63.1 | 62.9* |
| | | NMI | 63.4 | 64.5* | 64.9 | 63.7* | 64.1 | 63.9* | 64.8 | 62.9* |
| | MIE$^+$ | ACC | 66.3 | 66.4* | 66.8 | 66.7* | 66.6 | 66.4* | 67.2 | 67.2* |
| | | NMI | 65.8 | 65.8* | 66.0 | 66.0* | 65.8 | 65.8* | 65.6 | 65.6* |
| | Graph$^+$ | ACC | 67.8 | 67.2* | 67.4 | 67.5* | 67.1 | 67.2* | 67.5 | 67.8* |
| | | NMI | 67.4 | 67.2* | 66.7 | 67.1* | 66.9 | 67.3* | 66.5 | 66.1* |

objective functions are introduced in the supplementary material. We construct the original weight matrix $\mathbf{W}$ with probabilistic $K$-nearest neighbors on each dataset, and the number of neighbors is set to 3. Finally, we integrate the proposed adversarial learning strategy with some typical clustering frameworks, including variational deep embedding (VaDE) [14], deep spectral clustering using dual autoencoder network (DANDSC) [37], and improved deep embedding clustering (IDEC) [11], to further verify the performance of adversarial learning.

## 5.5 Results and Discussion

Firstly, we verify the performance of the proposed attack learning algorithm on different clustering networks. The results are shown in Table 1, where the first column represents the clustering performance of the clean features, the second column represents the clustering performance of using the perturbed features, and D represents the intensity of perturbation. The results show that the methods based on mutual information estimation are susceptible to perturbations because the distribution of the embedded features is more discrete. The graph module can effectively improve the clustering performance, but it is still easily affected by perturbations. The results imply that the robustness of the clustering network is independent of the clustering performance. In addition, the perturbation intensity decreases with the increasing of feature dimension, which indicates that the stability of the network is weakened with more complex feature structure.

In addition, we attempt to adopt the defense strategy to improve the network robustness using the generated perturbation. The results after defense optimization are shown in Table 2, where the first column represents the clustering performance of clean features and the second column represents the clustering performance of perturbed features. The results demonstrate that the proposed defense strategy can effectively mitigate the perturbation impact and even improve the overall clustering performance. The main reason for this phenomenon is that the clustering network will force the embedding space to have a clearer cluster structure during the defense process. Figure 3 is the feature distribution of the embedding space using $t$-SNE visualization [19] on MNIST-*test* data points, where

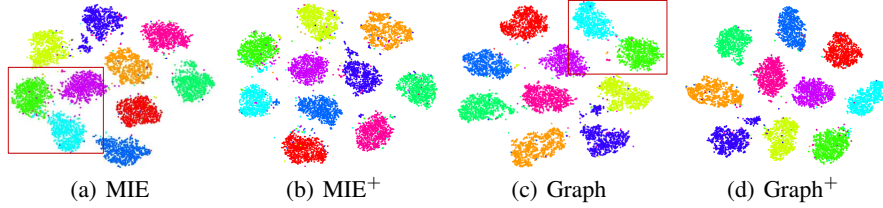

(a) MIE       (b) MIE$^+$       (c) Graph       (d) Graph$^+$

Figure 3: Visualization of the discriminative capability of the embedding spaces on MNIST-*test*.

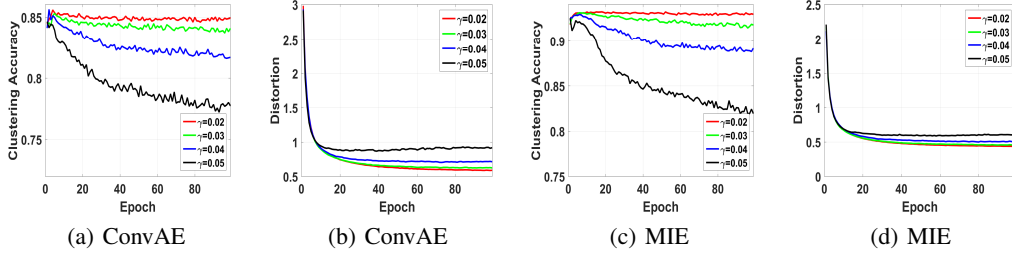

(a) ConvAE       (b) ConvAE       (c) MIE       (d) MIE

Figure 4: The clustering results of different clustering methods on MNIST during the attack strategy.

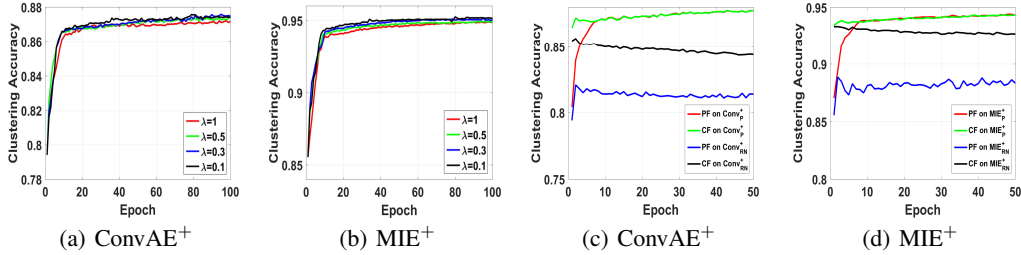

(a) ConvAE$^+$       (b) MIE$^+$       (c) ConvAE$^+$       (d) MIE$^+$

Figure 5: The clustering results of different clustering methods on MNIST during the defense strategy.

Table 3: Clustering performances (%) of different methods after the re-attack strategy on two datasets in ACC, NMI, and D.

| Method | Matrix | MNIST | | | | Fashion-MNIST | | | |
|---|---|---|---|---|---|---|---|---|---|
| | | 64 | 128 | 256 | 512 | 64 | 128 | 256 | 512 |
| ConvAE$^+$ | ACC | 80.2* | 81.2* | 82.0* | 81.2* | 58.9* | 57.8* | 58.3* | 59.2 * |
| | NMI | 75.4* | 75.7* | 75.2* | 74.3* | 60.3* | 59.7* | 60.7* | 61.4 * |
| | D | 1.52 | 1.25 | 1.71 | 1.87 | 2.15 | 2.57 | 3.01 | 2.88 |
| MIE$^+$ | ACC | 84.5* | 85.3* | 85.8* | 84.9* | 60.3* | 61.2* | 60.8* | 61.5* |
| | NMI | 80.1 * | 81.2 * | 82.1* | 81.7* | 60.5* | 61.7* | 60.4* | 61.9* |
| | D | 1.66 | 1.44 | 1.65 | 1.67 | 2.57 | 2.13 | 2.45 | 2.35 |
| Graph$^+$ | ACC | 92.1 * | 93.2* | 93.7 * | 92.8 * | 65.4* | 64.9* | 65.5* | 65.4* |
| | NMI | 86.7 * | 87.4 * | 88.2 * | 87.5 * | 60.6* | 60.3* | 61.6* | 61.9* |
| | D | 1.85 | 1.77 | 1.79 | 1.74 | 3.15 | 2.87 | 2.95 | 2.62 |

Table 4: Clustering performances (%) of different methods on two datasets in ACC and NMI.

| Dataset | IDEC | | IDEC$^+$ | | VaDE | | VaDE$^+$ | | DANDSC | | DANDSC$^+$ | |
|---|---|---|---|---|---|---|---|---|---|---|---|---|
| | ACC | NMI | ACC | NMI | ACC | NMI | ACC | NMI | ACC | NMI | ACC | NMI |
| MNIST | 87.4 | 82.3 | 95.2 | 94.1 | 94.3 | 88.2 | 96.7 | 91.3 | 97.5 | 93.2 | 98.1 | 94.0 |
| | 77.5* | 75.8* | 95.2* | 95.4* | 88.7* | 83.2* | 96.9* | 90.8* | 94.3* | 88.5* | 97.5* | 93.2* |
| Fashion-MNIST | 58.3 | 55.7 | 60.3 | 60.1 | 61.3 | 55.1 | 62.3 | 62.1 | 67.4 | 65.7 | 68.7 | 68.1 |
| | 59.8* | 56.2* | 61.7* | 61.8* | 62.4* | 56.5* | 62.9* | 63.2* | 66.7* | 60.8* | 67.1* | 67.5* |

Figures 3(a) and 3(c) are the distributions of the features from the original network with MIE and graph module, respectively, and Figures 3(b) and 3(d) are the feature distributions after the defense strategy. The results demonstrate that the features have clearer cluster structure for clustering after defense training, which can also be justified by the increased clustering accuracy.

We set $\beta = 1$. Figures 4(a) and 4(c) represent the change of clustering accuracies on adversarial samples with different $\gamma$. Figures 4(b) and 4(d) show the corresponding distortions on the MNIST

dataset, which indicate that the clustering accuracy will rapidly decay as the distortion increases gradually. We also investigate the parameter sensitivity in the proposed defense strategy. Figures 5(a) and 5(b) display the changes of clustering accuracies on perturbed features with various $\lambda$ values for ConvAE$^+$ and MIE$^+$, respectively. They both indicate that our defense strategy is insensitive to the parameter $\lambda$ in the range of [0.1,1].

To further prove the effectiveness of the adversarial samples, we adopt the learned perturbation (P) and random noise (RN) as the adversarial samples in the defense strategy, and verify the performance with clean features (CF) and perturbed features (PF), respectively, on the MNIST dataset. The results shown in Figures 5(c) and 5(d) demonstrate that the network quickly adapts to the perturbation and then iterates gradually to achieve better results. In particular, the perturbation information based on the adversarial attack algorithm is more helpful to improve the robustness and overall clustering performance compared against random noise, which is mainly due to the fact that the learned perturbation can explicitly explore unstable samples in the embedding space.

Moreover, we adopt the same parameters to re-attack the optimized network, and the results are shown in Table 3. The results indicate that the differences caused by the perturbations are significantly reduced, and the required perturbation intensity is also increased. The results confirm that the proposed algorithm can improve the robustness of the clustering network. Finally, we verify the flexibility of our proposed algorithm by combining the proposed adversarial learning method with some classic deep clustering frameworks. The results shown in Table 4 demonstrate that the proposed algorithm can be applied to different clustering frameworks and improve their robustness. Due to the space limit, please refer to the supplementary material for more experimental results and analyses.

# 6 Conclusion

In this paper, we first attempted to incorporate adversarial learning into deep clustering networks to enhance the clustering robustness. To achieve this goal, we defined an adversarial sample in the embedding space for deep clustering, and then proposed a powerful adversarial attack algorithm to learn a small perturbation which can fool the clustering layers but not impact the deep embedding. As such, unstable samples can be explored explicitly in the embedding space. In addition, we provided a simple yet efficient defense algorithm to promote clearer cluster structure and improve the robustness of the clustering network. Moreover, we integrated the adversarial learning algorithm with different clustering modules and multiple existing clustering frameworks, and the experimental results demonstrate that our adversarial method can improve their robustness effectively. Particularly, the proposed defense algorithm can further boost the overall performance of deep clustering networks in most cases.

## Acknowledgments

Our work was supported in part by the National Natural Science Foundation of China under Grant 62071361, the National Key R&D Program of China under Grant 2017YFE0104100 and 2020AAA0107600. Junchi Yan was sponsored by CCF-Tencent Open Fund RAGR20200113 and Tencent AI Lab Rhino-Bird Visiting Scholars Program.

## Broader Impact

As an important tool for unsupervised learning, deep clustering can be applied to big data analytics and statistics. However, the robustness and stability of a certain clustering network are prone to being attenuated since the labels are absent. The adversarial learning perspective, which can precisely attack network weaknesses, has played a significant role in malware detection and computer security. The proposed method in this work adopts adversarial learning to detect unstable samples and improve the robustness of deep clustering networks. The method can further improve the effectiveness of clustering algorithms in practical applications and reduce the dependence of deep learning on massive labeled data. Moreover, deep clustering networks are more sensitive to perturbations. The attack and defense strategies for clustering networks can improve network security and prevent malicious distorted information. However, uncontrolled applications in big data analytics and statistics may cause problems concerning users' information security and personal privacy.

## Footnotes

*The corresponding author.

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
