[Supplementary Material]

# Supplementary Material for
# "Adversarial Learning for Robust Deep Clustering"

**Xu Yang[1]**   **Cheng Deng[1]**   **Kun Wei[1]**   **Junchi Yan[2]**   **Wei Liu[3]**

[1]School of Electronic Engineering, Xidian University, Xian 710071, China
[2]Department of CSE and MoE Key Lab of Artificial Intelligence, Shanghai Jiao Tong University
[3]Tencent AI Lab, Shenzhen, China

`{xuyang.xd, chdeng.xd, weikunsk}@gmail.com, yanjunchi@sjtu.edu.cn, wl2223@columbia.edu`

This supplementary material includes two sections *i.e.*, details of baselines and descriptions of experiments.

## 1   Baselines

We adopt the following clustering network as a testbed to show how our proposed adversarial learning algorithm can attack the network and improve its robustness. The ultimate clustering network combines embedding and clustering as a whole to produce the optimal nonlinear embedding. Two modules are merged into one unified framework and jointly optimized by relative-entropy (equivalent to $KL$ divergence) minimization, and the loss function can be defined as:

$$KL(p(\mathbf{x}, \mathbf{z}, \mathbf{y})||q(\mathbf{x}, \mathbf{z}, \mathbf{y})) = \sum_{\mathbf{y}} \iint p(\mathbf{z}, \mathbf{y}|\mathbf{x}) p_{data}(\mathbf{x}) \ln \frac{p(\mathbf{z}, \mathbf{y}|\mathbf{x}) p_{data}(\mathbf{x})}{q(\mathbf{x}|\mathbf{z}, \mathbf{y}) q(\mathbf{z}, \mathbf{y})} \mathrm{d}\mathbf{z}\mathrm{d}\mathbf{x}. \quad (1)$$

With the generative model and the factorized joint probabilities, the loss of the clustering network can be calculated as:

$$\begin{aligned} \min_{\Theta} \ \mathcal{L}_C &= \sum_{\mathbf{y}} \iint p(\mathbf{z}, \mathbf{y}|\mathbf{x}) p_{data}(\mathbf{x}) \ln \frac{p(\mathbf{z}, \mathbf{y}|\mathbf{x}) p_{data}(\mathbf{x})}{q(\mathbf{x}|\mathbf{z}, \mathbf{y}) q(\mathbf{z}, \mathbf{y})} \mathrm{d}\mathbf{z}\mathrm{d}\mathbf{x} \\ &= \sum_{\mathbf{y}} \iint p(\mathbf{y}|\mathbf{z}) p(\mathbf{z}|\mathbf{x}) p_{data}(\mathbf{x}) \ln \frac{p(\mathbf{y}|\mathbf{z}) p(\mathbf{z}|\mathbf{x}) p_{data}(\mathbf{x})}{q(\mathbf{x}|\mathbf{z}) q(\mathbf{z}|\mathbf{y}) q(\mathbf{y})} \mathrm{d}\mathbf{z}\mathrm{d}\mathbf{x}. \end{aligned} \quad (2)$$

Equivalently, the loss of the clustering network becomes

$$\min_{\Theta} \mathcal{L}_C = \mathbb{E}_{\mathbf{x} \sim p_{data}(\mathbf{x})} \left[ -\log q(\mathbf{x}|\mathbf{z}) + \sum_{\mathbf{y}} p(\mathbf{y}|\mathbf{z}) \log \frac{p(\mathbf{z}|\mathbf{x})}{q(\mathbf{z}|\mathbf{y})} + KL(p(\mathbf{y}|\mathbf{z})||q(\mathbf{y})) \right], \mathbf{z} \sim p(\mathbf{z}|\mathbf{x}). \quad (3)$$

We intend to verify the robustness and stability of the basic clustering network mentioned above (ConvAE in the experiments). In addition, we integrate some classic modules of unsupervised learning with the basic clustering network to verify the applicability of the proposed adversarial algorithm, such as mutual information estimation (MIE) and graph module (Graph). Mutual information measures the essential correlation between two samples and can effectively estimate the similarity between features $\mathbf{z}$ and input data $\mathbf{x}$. The objective function of MIE module utilized in the experiments is defined as:

$$\begin{aligned} \mathcal{L}_C = \mathbb{E}_{\mathbf{x} \sim p_{data}(\mathbf{x})} \big[ &-\log q(\mathbf{x}|\mathbf{z}) + \sum_{\mathbf{y}} p(\mathbf{y}|\mathbf{z}) \log \frac{p(\mathbf{z}|\mathbf{x})}{q(\mathbf{z}|\mathbf{y})} \\ &+ KL(p(\mathbf{y}|\mathbf{z})||q(\mathbf{y})) - \log \frac{p(\mathbf{z}|\mathbf{x})}{p(\mathbf{z})} \big], \mathbf{z} \sim p(\mathbf{z}|\mathbf{x}), \end{aligned} \quad (4)$$

where $p(\mathbf{z}) = \int p(\mathbf{z}|\mathbf{x})p_{data}(\mathbf{x})\mathrm{d}\mathbf{x}$. Similarly, the objective function of Graph module is defined as:

$$\mathcal{L}_C = \mathbb{E}_{\mathbf{x}\sim p_{data}(\mathbf{x})} \left[ -\log q(\mathbf{x}|\mathbf{z}) + \sum_{\mathbf{y}} p(\mathbf{y}|\mathbf{z}) \log \frac{p(\mathbf{z}|\mathbf{x})}{q(\mathbf{z}|\mathbf{y})} + KL(p(\mathbf{y}|\mathbf{z})||q(\mathbf{y})) \right] \\ + \frac{1}{n^2} \sum_{i,j=1}^{n} \mathbf{W}_{i,j} \parallel \mathbf{y}_i - \mathbf{y}_j \parallel^2, \tag{5}$$

where $\mathbf{W}$ is the weight matrix with probabilistic $K$-nearest neighbors on each dataset, and the number of neighbors is set to 3.

## 2  Experimental Results

To further show that our method operates well on various datasets, we choose STL-10 [1] and REUTERS [3] as additional benchmarks. STL-10: The STL-10 dataset consists of color images of $96 \times 96$ pixel size, in which there are 10 classes with 1,300 examples each. Following [2], we extract features of images of STL-10 by ResNet-50, which are then used to test the performance of all baselines. REUTERS: A random subset of 10,000 documents with 4 root categories: corporate/industrial, government/social, markets, and economics as labels are used. We compute *tf-idf* features on the 2,000 most frequent words to represent all articles. The results are shown in Tables 1, 2, and 3, which give the performances of attack, defense, and re-attack, respectively. Such results demonstrate that the proposed adversarial learning method can be readily applied to a variety of datasets and achieve consistently competitive performances.

Table 1: Clustering performances (%) of different methods after adversarial attack learning on two datasets in ACC, NMI, and D.

| Dataset | Method | Matrix | 64 | | 128 | | 256 | | 512 | |
|---------|--------|--------|------|------|------|------|------|------|------|------|
| STL-10 | ConvAE | ACC | 84.7 | 78.1* | 81.2 | 75.6* | 81.9 | 73.3* | 80.6 | 65.8* |
| | | NMI | 86.2 | 80.2* | 84.4 | 77.9* | 83.6 | 75.7* | 80.6 | 72.1* |
| | | D | 2.77 | | 2.48 | | 0.74 | | 0.40 | |
| | MIE | ACC | 86.7 | 80.2* | 85.2 | 79.5* | 84.7 | 78.7* | 84.9 | 79.5* |
| | | NMI | 85.4 | 79.5* | 84.4 | 80.4* | 84.5 | 79.5* | 84.2 | 80.1* |
| | | D | 2.44 | | 1.95 | | 1.06 | | 0.57 | |
| | Graph | ACC | 92.5 | 80.0* | 91.0 | 84.6* | 91.1 | 84.8* | 90.6 | 82.5* |
| | | NMI | 87.6 | 77.7* | 86.5 | 80.0* | 86.6 | 79.8* | 86.0 | 78.6* |
| | | D | 5.14 | | 4.67 | | 4.01 | | 4.24 | |
| REUTERS | ConvAE | ACC | 75.3 | 70.6* | 74.3 | 69.2* | 74.7 | 69.1* | 73.9 | 68.2* |
| | | NMI | 50.0 | 44.5* | 49.2 | 44.8* | 50.2 | 45.7* | 51.2 | 44.8* |
| | | D | 3.13 | | 2.75 | | 2.83 | | 2.12 | |
| | MIE | ACC | 74.2 | 68.3* | 73.7 | 67.5* | 76.3 | 69.3* | 74.2 | 67.7* |
| | | NMI | 49.2 | 42.1* | 48.5 | 41.7* | 50.8 | 43.2* | 49.1 | 42.5* |
| | | D | 2.57 | | 2.35 | | 2.17 | | 2.15 | |
| | Graph | ACC | 80.8 | 72.0* | 80.4 | 71.9* | 79.5 | 71.8* | 80.9 | 72.1* |
| | | NMI | 57.8 | 39.6* | 57.4 | 40.1* | 56.7 | 39.4* | 58.4 | 39.4* |
| | | D | 3.15 | | 3.03 | | 2.87 | | 2.74 | |

We select some samples with inconsistent clustering results before and after the attack strategy. The results with different $\gamma$ values are shown in Figure 1, where Figure 1(a) are original samples, Figure 1(b) are generated samples by clean features, and Figures 1(c)-1(f) are generated samples from the perturbed features with different $\gamma$. The images generated by these features are visually very similar but the corresponding clustering results are quite different, which indicates that the learned perturbation can easily fool the clustering layers but not impact the performance of the deep embedding.

## References

[1] Adam Coates, Andrew Ng, and Honglak Lee. An analysis of single-layer networks in unsupervised feature learning. In *Proceedings of the fourteenth international conference on artificial intelligence and statistics*, pages 215–223, 2011.

Table 2: Clustering performances (%) of different methods after the adversarial defense strategy on two datasets in ACC and NMI.

| Dataset | Method | Matrix | 64 | | 128 | | 256 | | 512 | |
|---|---|---|---|---|---|---|---|---|---|---|
| STL-10 | ConvAE$^+$ | ACC | 87.4 | 87.6* | 85.2 | 85.0* | 84.8 | 84.3* | 85.1 | 84.9* |
| | | NMI | 86.5 | 86.3* | 85.4 | 85.9* | 85.1 | 85.6* | 84.2 | 84.0* |
| | MIE$^+$ | ACC | 89.5 | 89.2* | 90.2 | 90.3* | 90.7 | 90.1* | 89.9 | 89.5* |
| | | NMI | 87.2 | 87.0* | 86.6 | 86.9* | 86.2 | 86.0* | 85.8 | 85.3* |
| | Graph | ACC | 94.7 | 94.5* | 95.1 | 94.8* | 94.5 | 94.3* | 95.2 | 94.9* |
| | | NMI | 89.5 | 89.2* | 89.7 | 88.7* | 88.2 | 88.1* | 88.5 | 88.3* |
| REUTERS | ConvAE$^+$ | ACC | 77.6 | 77.5* | 76.8 | 76.5* | 78.3 | 78.5* | 77.6 | 77.3* |
| | | NMI | 53.2 | 53.1* | 52.9 | 52.7* | 53.1 | 53.9* | 52.8 | 52.2* |
| | MIE$^+$ | ACC | 75.5 | 75.4* | 76.1 | 75.7* | 76.3 | 76.2* | 75.7 | 75.9* |
| | | NMI | 50.3 | 50.2* | 51.2 | 51.0* | 50.8 | 50.9* | 49.7 | 49.6* |
| | Graph$^+$ | ACC | 82.3 | 82.3* | 83.1 | 83.2* | 83.3 | 83.1* | 82.7 | 82.8* |
| | | NMI | 58.9 | 58.2* | 57.4 | 58.1* | 58.3 | 57.9* | 58.0 | 57.5* |

Table 3: Clustering performances (%) of different methods after the re-attack strategy on two datasets in ACC, NMI, and D.

| Method | Matrix | STL-10 | | | | REUTERS | | | |
|---|---|---|---|---|---|---|---|---|---|
| | | 64 | 128 | 256 | 512 | 64 | 128 | 256 | 512 |
| ConvAE$^+$ | ACC | 82.1* | 79.5* | 81.7* | 81.5* | 73.5* | 74.8* | 73.7* | 74.1 * |
| | NMI | 84.3* | 80.7* | 80.9* | 81.2* | 48.7* | 48.9* | 47.6* | 48.2 * |
| | D | 4.13 | 3.75 | 2.53 | 2.37 | 4.25 | 4.32 | 3.87 | 3.57 |
| MIE$^+$ | ACC | 85.2* | 85.1* | 84.7* | 84.5* | 71.2* | 70.4* | 70.5* | 71.2* |
| | NMI | 83.7 * | 83.8 * | 83.4* | 83.7* | 51.2* | 50.5* | 50.8* | 51.4* |
| | D | 3.66 | 2.56 | 2.04 | 2.17 | 3.17 | 2.89 | 2.57 | 2.30 |
| Graph$^+$ | ACC | 90.5 * | 90.4* | 89.7 * | 89.3 * | 75.7* | 74.8* | 75.2* | 74.7* |
| | NMI | 86.7 * | 86.5 * | 87.1 * | 87.5 * | 50.2* | 49.8* | 50.6* | 48.9* |
| | D | 7.01 | 6.57 | 6.32 | 5.92 | 4.23 | 4.12 | 3.89 | 3.19 |

(a) Data samples  (b) Generated clean samples  (c) $\gamma$=0.02

(d) $\gamma$=0.03  (e) $\gamma$=0.04  (f) $\gamma$=0.05

Figure 1: Some clustering samples of MNIST. a) Data samples; b) The samples generated by clean features; c-f) The samples generated by perturbed features with different $\gamma$.

[2] Zhuxi Jiang, Yin Zheng, Huachun Tan, Bangsheng Tang, and Hanning Zhou. Variational deep embedding: An unsupervised and generative approach to clustering. *arXiv:1611.05148*, 2016.

[3] David D Lewis, Yiming Yang, Tony G Rose, and Fan Li. Rcv1: A new benchmark collection for text categorization research. *Journal of machine learning research*, 5(Apr):361–397, 2004.