[Reviews · NeurIPS 2020]

Review 1

Summary and Contributions: The paper is devoted to improving the robustness of deep clustering algorithms by adversarial learning. To achieve this, the authors propose a new adversarial attack strategy and a corresponding defense algorithm. Experiments support their claim and well validate the effectiveness of the proposed method. This method is proved to be able to improve the robustness and performance of some existing deep clustering algorithms. The idea and solution can make important contributions to and have potential impact on the deep clustering community.

Strengths: 1.The problem of the robustness of deep clustering is important but overlooked by existing works. This paper is the first work to solve this problem by using adversarial learning. I hope more works follow this direction and try to solve the problem in different aspects. 2.The solution is intuitive and effective. They propose a new adversarial attack strategy that can determine unreliable samples for clustering. Then they give a corresponding defense algorithm that can make bad adversarial samples normal again. As a result, the final network can resist the adversarial attack and make the cluster structure clearer. 3.Extensive experiments well validate the effectiveness of the proposed method. 2) The authors have given a thorough review and comparison on the definitions of most existing MI estimation methods. 3) Extensive experiments well validate the effectiveness of the proposed method.

Weaknesses: Cons: 1.The methodology is not too clear. Which part of parameters do \Psi and \Phi represent in detail? Is \delta included in \Phi? How are T and \sigma in line#130 defined and computed? Why is the last term $\sum y_i y_i^\top$ is included in the expectation E_{x~q(x|z)}? 2.There are some grammar errors. Eg. “the features is” should be “the features are” in line#86. The sentence “For a pretrained deep clustering...” in line#109-111 does not have a subject. In line#112, “adversarial samples is defined...” should be “adversarial features are defined...”? Suggestion: Release the source code after the acceptance or add more implementation details for re-producing the experimental results.

Correctness: Yes

Clarity: Yes

Relation to Prior Work: Yes

Reproducibility: Yes

Additional Feedback:


Review 2

Summary and Contributions: This paper introduces a probabilistic framework to make deep clustering approaches more robust to adversarial attacks. The main contribution of the paper consists in proposing a loss to jointly train a deep clustering model, an attacker and a defender for that purpose. The framework is validated on MNIST and Fashion-MNIST (and 2 more datasets in the supplementary material). I have read the rebuttal and I appreciate the efforts for clarification. However, there are still important points that remain unclear or that were not covered in the rebuttal: - The link between the proposed formalization in Sec. 2 and that from [30] is still not clear and requires more details for the reader to understand. - The rebuttal did not address the points I raised about the loss in Eq. 3 being ill-defined. Therefore, I am keeping my score.

Strengths: - The experiment results show that the proposed defense strategy effectively counters the perturbations and maintains the performance in the perturbed setting close to or even better than in the unperturbed setting. - The proposed attack/defense strategies seem flexible and can be adopted for improving the robustness of different deep clustering approaches.

Weaknesses: - The description of the framework and the writing is not sufficiently clear; it lacks important details for the reader to understand and reproduce the work. - The formulation of the clustering loss does not seem sound. - Novelty is fairly limited as the work mostly consists in applying adversarial learning to deep clustering.

Correctness: The formulation which yields the clustering loss, described in Section 2, does not seem correct. The clustering loss is defined as the Kullback-Leibler divergence between p(x, z, y) and q(x, z, y). The expectation over x drawn from p_data is applied to the KL divergence. It thus means that the KL itself does not contain an expectation over x (but only over z and y) and x in the KL is actually only a draw x0 of the random variable. However, p(x=x0, z, y) and q(x=x0, z, y) are not probability distributions on (z, y) and using these quantities in a KL divergence is therefore not appropriate. Furthermore, I did not understand how Equation (3) is derived from Equation (1). The quantity q(x|z) contains an unbound z, on which no expectation is taken. Additionally, out of symmetry, I would have expected the 3rd term in the expectation to be p(z|x) * KL(p(y|z) || q(y)), as otherwise the quantity p(z|x) that appears in the second term is missing. In terms of empirical methodology, it is unclear how the hyperparameters beta and gamma (which are claimed to be specified for each dataset) are determined in this unsupervised setting, without any labeled data.

Clarity: The paper misses important details which makes some parts of the paper difficult to understand. Additionally, the clarity and quality of the writing could be improved; thorough proofreading is needed. Many notations were not introduced or are unclear: - There is no mention of what p and q exactly are in Section 2. It also seems to differ from the convention found in some papers that p is the true distribution and q is the variational distribution (e.g., in [12]). It is therefore expected that the roles of those quantities are clearly defined. - Is the learned perturbation delta the same for all samples? - What is T in Section 3? How is it defined? - The notations for the random variables (e.g., x ~ Bernoulli(...)) and the samples (e.g., x = {x_1, ..., x_n}) are merged which may confuse the reader and makes it harder to understand the formulation. Some experimental details also need clarifications: - What does the method denoted as "Conv" refer to? MIE and Graph were introduced in Section 5.4, but I could not find any description of Conv. - What is the difference between Table 2 and Table 3? The caption is almost exactly the same and it is not easy to understand from the text how the setup differs between these two tables. - In Table 4, the defense strategy is applied to different algorithms (IDEC, VaDE and DANDSC). However, it is not straightforward how the proposed framework can be applied to any deep clustering approach -- in particular, to a non-probabilistic approach such as IDEC.

Relation to Prior Work: It is not clear if the probabilistic formulation of deep clustering introduced in Section 2 is a contribution of this paper or if it comes from previous works. There is important similitude with the formulation from VaDE [12] but this work is not cited in this section. The authors should make the link to this work more explicit.

Reproducibility: No

Additional Feedback: In the example of Section 1 to motivate the need for defense against adversarial attacks in deep clustering, I would have appreciated a more quantitative perspective. It is only mentioned that "a small perturbation will cause the clustering results to be quite different", without giving explicit numbers about how impactful the perturbation is. The "broader impact" section mentions that the proposed approach to make deep clustering models robust to adversarial attacks could be useful for network security and against "malicious distorted information". More details on the link between clustering and network security would be helpful for the reader.


Review 3

Summary and Contributions: I read the rebuttal and would like keep my score. the authors are encouraged to revise the paper by adopting the comments from the reviewers. This paper proposes a robust deep clustering method based on adversarial learning. The adversarial attack in the embedding subspace is utilized to explore the samples that easily fool the clustering layers, and the defense algorithm is proposed with the learned adversarial samples to improve the robustness of clustering.

Strengths: 1. Clustering remains an important unsupervised learning tool in many applications. However, a small perturbation will cause the clustering results to be quite different, and the reconstruction loss of the embedding network cannot perceive these perturbations well. The paper adopts the adversarial attack to learn the samples that easily fool the clustering layers and not impact the performance of the embedding network. The motivation is interesting and the idea seems novel. 2. The proposed adversarial samples are utilized to improve the robustness of clustering results. Compared to traditional random noise, this learned noise is obviously more conducive to improving the performance of the network. 3. The model provides a complete attack and defense strategy and effectively solves the robustness problem of the embedded space. 4. The results demonstrate that the proposed method is generally applicable to multiple existing clustering frameworks and clustering modules to boost their robustness. The results show that the network has considerable applicability.

Weaknesses: 1. To further facilitate reading and understanding, the author should provide specific objective functions when the model is combined with different clustering modules in the text or supplementary materials. 2. In the supplementary material, the author shows the difference between the samples generated by the disturbance features. In order to better understand the difference, the author should provide specific indicators to quantify this difference compared with the original samples. 3. Some detailed descriptions need to be provided. In Figure 2, how the generated images and their corresponding y are combined?

Correctness: yes

Clarity: yes

Relation to Prior Work: yes

Reproducibility: Yes

Additional Feedback: 1. To facilitate reading and understanding, the author should provide specific objective functions when the model is combined with different clustering modules in the text or supplementary materials. 2. In order to better understand the difference between the samples generated by the disturbance features and clean features, the author should provide specific indicators to quantify this difference compared with original samples. 3. The author needs to add some detailed descriptions. In Figure 2, how the generated images and their corresponding y are combined?

[Author Response · NeurIPS 2020]

First of all, we would like to thank all the reviewers' valuable comments and their recognition (mainly from R #3 and R #5) on the contributions of our work in terms of strong motivation, technical originality, wide applicability to various clustering frameworks, and convincing experiments with a comprehensive ablation study. Our response is as follows, which mainly addresses the concerns and perhaps some potential misunderstandings of R #4. Our source code will be released and more details will be added by the 9th extra page in the revised version. We understand that the reviewers may have missed some critical points in such a short review period, while we do hope that R #4 could reconsider the contributions and novelty of our work based on our clarification.

## 1    To Reviewer 3

**Clarity**. Thanks for your questions. $\Phi$ and $\Psi$ denote the parameters of the attack network and the discriminator respectively, and $\delta$ is the output of the attack network. Notation $T$ is the mapping function of the discriminator (line 128-130) constructed by the neural network, which is often used to discriminate the relevances of the inputs [11], and $\sigma$ represents the activation function of the discriminator. Negative sampling estimation [11] is adopted to optimize the mapping function $T$. We will improve the writing and release the source code.

## 2    To Reviewer 4

**Understanding Eq. 1 and Eq. 3**. These equations refer to the clustering model introduced in Section 2, which is a basic model of an existing method [30] used as a testbed to show how our proposed adversarial algorithm can attack the network and improve its robustness, and our technique can be widely applied to more existing clustering methods which are essentially attack-agnostic. The loss function is $\mathcal{L}_C = KL(p(\mathbf{x}, \mathbf{z}, \mathbf{y}) || q(\mathbf{x}, \mathbf{z}, \mathbf{y}))$, and Eq. 3 is an approximate loss function of the model in the actual implementation which aims to indicate the physical meaning of the model. The first and third terms represent the reconstruction loss and the satisfied distribution of $\mathbf{y}$, respectively. The second item is utilized to make $\mathbf{z}$ align with the 'exclusive' distribution of a certain cluster. Due to the space limit, we refrain ourselves from expanding too many details of this method, which is not our technical purpose. Certainly, we will proofread carefully and add more details in the revised version given additional one page to improve the readability.

**Its applicability to other deep clustering networks**. The reviewer has expressed the concern about how our technique can be applied to other clustering networks, as it is supposed to be coupled with the clustering model presented in Section 2 (while in fact NOT). Our adversarial learning algorithm is **model-agnostic** and can therefore be applied to the deep clustering model that follows the $\mathbf{x} \rightleftarrows \mathbf{z} \rightarrow \mathbf{y}$ structure. The model in Section 2 (the Conv in the experiments) is one of such models leveraged to introduce one adversarial learning algorithm and afterwards verify its effectiveness encountering an attack algorithm. $\mathcal{L}_C$ changes as the clustering model changes. For other deep clustering models, e.g., VaDE and IDEC, $\mathcal{L}_C$ is their corresponding objective function. To be precise, our defense algorithm is to integrate a set of perturbation-based contrastive constraints into the original clustering model, which can use the learned perturbations to improve both the clustering performance and robustness. Particularly, VaDE [12] assumes a mean-field approximation in the generative process, which is different from the model introduced in Section 2.

**Hyper-parameter test**. We aim to select the hyper-parameters that can achieve a certain difference in the result (the 3rd term of Eq.6) by a moderate perturbation (not too much). The roles of $\gamma$ and $\beta$ are mutually exclusive, so we typically fix $\beta$ and tune $\gamma$. The experiments show that the difference in the result will increase abruptly as $\gamma$ gradually increases, and the critical value $\gamma$ is an ideal hyper-parameter. For example, the difference in the result (not clustering accuracy) is 0.05 (from 0.88 to 0.83) when $\gamma$=0.05 for the Conv with 128-D features, while the difference is 0.21 (from 0.88 to 0.67) when $\gamma$=0.06. Obviously, $\gamma$=0.05 is relatively suitable, and the final clustering result also verifies this choice. The clustering accuracy drops to 0.592 when $\gamma$=0.06, which has almost killed the basic capability of the network.

**Novelty and contributions**. We re-summarize our contributions in the context of R #4's concerns in three aspects: **1)** To the best of our knowledge, this paper is the first work for learning unsupervised adversarial clustering networks, which is in fact very important due to the vulnerability nature of both deep clustering and unsupervised learning (see the comments from R#3 and R#5) but has rarely been studied in the literature. Our research may inspire more efforts from the machine learning community. **2)** We propose both novel attack and defense strategies which can be readily applied to most existing deep clustering methods, owing to our technical design. Our attack technique can also be used to identify the unreliable samples from unlabeled data, permitting data mining applications. **3)** We perform extensive experiments to corroborate the effectiveness of our method as well as the importance of adversarial learning for existing deep clustering models, through the three parts in the experiments on attack, defense, and re-attack strategies.

**Points for clarification**. We will try our best to improve the writing and release the code. Here are some specific explanations: **1)** The description of Table 3 is shown in line 236-239 in the main paper, where we re-attack the network after our defense strategy to further verify the effectiveness of the defense strategy. **2)** Please refer to our response to **Reviewer** #3 for the details about the discriminator $T$. **3)** The attack strategy will learn the corresponding perturbation according to each sample. **4)** The example of Section 1 is the results of the basic clustering model (Conv) with 128-D features. The clustering accuracy dropped from 0.849 to 0.772.

## 3    To Reviewer 5

**1)** For the MIE and Graph modules, their corresponding $\mathcal{L}_C$ adds mutual information $I(\mathbf{x}, \mathbf{z})$ and graph constraints $\sum_{i,j} \mathbf{W}_{ij} \parallel \mathbf{y}_i - \mathbf{y}_j \parallel_2^2$ to the basic model. Due to the space limit, we will add a complete formula in the revised version. **2)** The generated images and their corresponding $\mathbf{y}$ are combined via feature reshaping. The combined result and its latent representation $\mathbf{z}$ together form a sample pair as the input of the discriminator.

[Meta-Review · NeurIPS 2020]

This paper presents a framework to improve the robustness of deep clustering to adversarial attack. This problem is important, and the method is sound and backed by extensive experimentation. However, the paper is in part not entirely clear and hard to reproduce with the current level of details, and a major rewriting is required to clarify the details of this method.